# Flow Simulation and Gradient Printing of Fluorapatite- and Cell-Loaded Recombinant Spider Silk Hydrogels

**DOI:** 10.3390/biom12101413

**Published:** 2022-10-03

**Authors:** Vanessa J. Neubauer, Florian Hüter, Johannes Wittmann, Vanessa T. Trossmann, Claudia Kleinschrodt, Bettina Alber-Laukant, Frank Rieg, Thomas Scheibel

**Affiliations:** 1Lehrstuhl Biomaterialien, Fakultät für Ingenieurwissenschaften, Universität Bayreuth, Prof.-Rüdiger-Bormann-Straße 1, 95447 Bayreuth, Germany; 2Lehrstuhl Konstruktionslehre und CAD, Fakultät für Ingenieurwissenschaften, Universität Bayreuth, Universitätsstraße 30, 95440 Bayreuth, Germany; 3Bayreuth Engine Research Center (BERC), Universität Bayreuth, Universitätsstraße 30, 95440 Bayreuth, Germany; 4Zentrum für Energietechnik (ZET), Universität Bayreuth, Universitätsstraße 30, 95440 Bayreuth, Germany; 5Bayreuther Zentrum für Kolloide und Grenzflächen (BZKG), Universität Bayreuth, Universitätsstraße 30, 95440 Bayreuth, Germany; 6Bayerisches Polymerinstitut (BPI), Universitätsstraße 30, 95440 Bayreuth, Germany; 7Bayreuther Zentrum für Molekulare Biowissenschaften (BZMB), Universität Bayreuth, Universitätsstraße 30, 95440 Bayreuth, Germany; 8Bayreuther Materialzentrum (BayMAT), Universität Bayreuth, Universitätsstraße 30, 95440 Bayreuth, Germany

**Keywords:** gradient printing, bioprinting, computational fluid dynamics, recombinant spider silk hydrogels, apatite particles, single cartridge set-up, tissue engineering

## Abstract

Hierarchical structures are abundant in almost all tissues of the human body. Therefore, it is highly important for tissue engineering approaches to mimic such structures if a gain of function of the new tissue is intended. Here, the hierarchical structures of the so-called enthesis, a gradient tissue located between tendon and bone, were in focus. Bridging the mechanical properties from soft to hard secures a perfect force transmission from the muscle to the skeleton upon locomotion. This study aimed at a novel method of bioprinting to generate gradient biomaterial constructs with a focus on the evaluation of the gradient printing process. First, a numerical approach was used to simulate gradient formation by computational flow as a prerequisite for experimental bioprinting of gradients. Then, hydrogels were printed in a single cartridge printing set-up to transfer the findings to biomedically relevant materials. First, composites of recombinant spider silk hydrogels with fluorapatite rods were used to generate mineralized gradients. Then, fibroblasts were encapsulated in the recombinant spider silk-fluorapatite hydrogels and gradually printed using unloaded spider silk hydrogels as the second component. Thereby, adjustable gradient features were achieved, and multimaterial constructs were generated. The process is suitable for the generation of gradient materials, e.g., for tissue engineering applications such as at the tendon/bone interface.

## 1. Introduction

The hierarchical organization of tissues is significantly contributing to their function [1]. Hierarchically arranged gradient materials often allow bridging two materials with different mechanical properties gaining smooth transitions and, therefore, preventing material failure at their interface [2]. This concept can be found, for example, at the tendon-bone insertion, the so-called enthesis, with material gradients and resulting in gradually changing mechanical properties [3]. Previously, biomechanics were investigated to determine threshold parameters for material design [4], and biomimetic gradient materials were generated for tendon-bone-interface reconstruction, e.g., using a gradual calcium phosphate coating on electrospun nanofiber meshes [5,6]. Another approach focused on the mechanical requirements of heterogeneously composed materials with changes in local stiffness [7].

A different approach for gradient processing is 3D printing, a versatile tool to generate predefined complex hierarchical structures. The simultaneous printing of two or more materials has so far been realized using co-printing approaches from separate containers resulting in multilayer constructs [8,9,10,11], concentration patterns [12,13], or from combined cartridges with a mixing unit [14,15] to generate scaffolds usable in tissue engineering approaches [16,17]. Importantly, such scaffolds have to be biocompatible, biodegradable and should show no toxicity or cause no immune reaction, limiting the choice of material [18].

To gain necessary mechanical features, filler materials can be implemented in soft hydrogel carriers to yield hard tissue engineering materials with increasing mechanical stiffness and further ones providing cell binding motives or chemical cues [19,20,21]. An interesting type of filler for hard tissue engineering is fluorapatite, containing fluoride instead of hydroxyl counterions in the apatite species. Fluoride is known as an essential trace element typically found in enamel and interconnected with dental applications [22,23]. However, fluoride is also required for bone formation from blood plasma and can induce mineralization processes in collagen or gelatine composites [24,25]. Fluorapatite (FAp, Ca_5_(PO_4_)_3_F) can be produced using precipitation methods [23] or ultrasonication [26,27]. FAp has already been successfully used for osteoporosis treatment [22] and bone tissue engineering [28] or as bioactive particles in composite scaffolds [29].

In this study, a combined numerical and experimental approach was used to generate gradient constructs using a novel printing method by printing out of one printer cartridge filled with two different material blocks. The focus was set on the evaluation of the bioprinting process concerning gradient generation. Computational fluid dynamics allowed the prediction of gradient formation using a transient laminar flow simulation in a commercial printer cartridge filled with an AB material block system taking into account the typical shear-thinning material behavior using Ostwald de Waele’s power law. The findings were transferred to an experimental approach using hydrogels made of the recombinant spider silk protein eADF4(C16). Recombinant spider silk proteins [30] have been previously established concerning (hydro)gel [31,32] formation for 3D printing of cells [33,34], particle [35,36] loaded constructs or material blends with gelatine [37]. Here, printed spider silk hydrogel gradients were obtained with increasing fluorapatite and cell (mouse fibroblasts) content, both to the same side of the construct. Our results confirm the possibility of generating particle- as well as cell-loaded gradient hydrogels out of one cartridge with controllable properties, which is a prerequisite for a later biofabrication of soft-to-hard tissue interfaces.

## 2. Materials and Methods

### 2.1. Computational Flow Simulation

ANSYS Version15 CFX was used to numerically simulate gradient formation, which follows the Finite Volume Method as a discretization technique [38]. Due to symmetry, the printer cartridge was simplified to a quarter model, as depicted in Appendix A. Only the fluid domain was modelled within the simulation set-up, where the inner wall of the real cartridge was mimicked by the outer contour of the fluid domain. The piston crown was approximated by a planar wall, as the distance to the outlet was large enough to neglect the influence of the real piston geometry. In addition, extrusion adapters, such as cannulas or cones, were not considered to simplify the model. The two phases have been designed as two separate blocks, here referred to as the AB block system, with a flat contact interface in between. The initial ratio of the lengths of blocks A and B was 1:2 to ensure a flow of almost only material B at the end of the gradual mixing process. Furthermore, this ensured that the mixing area was located far away from the piston so that the piston shape did not influence mixing. Both liquids A and B were modelled as homogenous non-Newtonian fluids. The power law according to Ostwald de Waele [39] was used for simulating the shear thinning properties of the recombinant spider silk eADF4(C16) hydrogel:(1)$$\tau =k{\dot{\gamma }}^{n}$$
where τ [Pa] denotes the shear stress and $\dot{\gamma \;}\left[s^{-1}\right]$ the shear rate. The flow consistency index was set to $k=148.89\;Pa*s^{n}$ with a flow behavior index of n=0.2025. The material parameters were fitted from measured rheological data within a range of $\dot{\gamma \;}=0.1-10\;s^{-1}.$ The boundary conditions were defined as depicted in Appendix A. Slip boundary conditions were applied to the outer boundaries of the cartridge, whereas free slip behavior was defined for the fluid interface of the domains and symmetry planes of the quarter model.

During the printing process, the piston crown moved towards the outlet, activating fluid flow motion. A displacement boundary condition was applied, causing the piston crown wall to move at a speed of v=5 mm/s. Since domain B was compressed during the printing process, a mesh motion condition was defined for domain B. The transient flow simulation was performed using a second-order backward Euler solution algorithm [40] with a time step size of Δt=0.1 s.

### 2.2. Bioprinting

A regenHU 3D Discovery Gen1 (Villaz-Saint-Pierre, Switzerland) bioplotter was used for bioprinting, equipped with cartridges size 3cc and regenHU conical pistons. Luer lock conical needles were adapted to the cartridges with an inner diameter of 14 G, 16 G or 20 G. The different materials were filled into the cartridges manually using block volumes of 0.5–1 mL. Polystyrene Petri dishes were used as substrate (diameter 8 cm, Sarstedt, Nümbrecht, Germany). A round-shaped monolayer construct was designed, and the according G-code was generated using the regenHU BioCAD V1.1 printer software. The applied pressure was set to 0.1 bar for all hydrogels. The printing speed was pre-set to 20 mm/s.

### 2.3. Recombinant Spider Silk Protein Production, Protein Labeling, and Hydrogel Preparation

The amino acid sequence of eADF4(C16) is based on the consensus sequence of the repetitive core domain of the dragline silk fibroin 4 of the European garden spider *Araneus diadematus*, the so-called C-module (GSSAAAAAAAASGPGGYGPENQGPSGPGGYGPGGP). The C-module is repeated 16 times to yield eADF4(C16). In eADF4(κ16), all glutamic acid residues are replaced by lysine ones. Recombinant spider silk proteins were produced and purified as previously described [41,42]. Hydrogels formed after protein dialysis and concentration adjustment by water removal using dialysis against PEG (Carl Roth, Karlsruhe, Germany) [31]. Covalent coupling of NHS-fluorescein (Waltham, MA, USA) to the amino-terminus of eADF4(C16) was conducted, as reported previously [31], with a 10-fold molar excess of dye.

### 2.4. Fluorescence Imaging and Fluorescence Spectroscopy

Fluorescence images of printed gradient recombinant spider silk hydrogels with and without fluorescent labeling were recorded at CY2 mode (Ex 480 nm/Em 530 nm, exposure 0.05) using an Ettan DIGE imager (GE Healthcare, Danderyd, Sweden).

Fluorescence spectra (n1 = 3 samples per spot on the scaffold; n2 = 3 scaffolds) were recorded using a fluorescence spectrometer FP-6300 (JASCO, Pfungstadt, Germany) and an excitation wavelength of 488 nm at 20 °C. Then, 3 µL of hydrogel samples were taken at equal distances (3 cm) from the printed strands and resuspended in 0.25 mL 10 mM Tris buffer, pH 7.5, before each measurement.

### 2.5. Fluorapatite Particle Synthesis

Briefly, 14.6 g calcium chloride (CaCl_2_, Carl Roth, Karlsruhe, Germany), 14.3 g disodium hydrogenphosphate (Na_2_HPO_4_, Carl Roth, Germany), and 0.8 g sodium fluoride (NaF, Carl Roth, Germany) were mixed in dry state, and 40 mL of MilliQ water were added shortly before ultrasonication [43] for 5 min at an energy intake of 18 kJ using a Sonoplus Ultrasonic Homogenizer (Bandelin, Berlin, Germany) and a KE76 probe. Particles were washed with MilliQ water and air dried at 50 °C overnight.

FAp particles were coated in an aqueous solution using eADF4(κ16) dissolved in 6 M guanidinium thiocyanate and dialyzed against 10 mM Tris buffer, pH 7.5 for 16 h. The coating was performed using 10 mg particles in 1 mg/mL protein solution for 8 h, followed by centrifugation at 13,000 rpm for 10 min and washing in MilliQ water. κFAp refers to FAp particles coated with eADF4(κ16).

### 2.6. Attenuated Total Reflectance-Fourier Transform Infrared Spectroscopy

Attenuated Total Reflectance-Fourier Transform Infrared (ATR-FTIR) spectra (n = 3) of particle species were recorded in a dry state using a germanium crystal mounted on a Bruker Tensor 27 (Ettlingen, Germany) at a resolution of 2 cm^−1^ using 100 scans. An atmospheric compensation algorithm was performed using OPUS 8.0 software to correct water vapor and carbon dioxide fluctuations during the measurements.

### 2.7. Microscopy

Fluorapatite particles were studied using SEM-imaging after carbon sputtering (20 nm) with a ZEISS Sigma 300 VP and Sigma 500 chamber equipped with an EDS detector (EDAX Pegasus and Octane Super Detector, 60 mm^2^ chip, Zeiss, Wetzlar, Germany) at an acceleration voltage of 7.5 kV to excite the Kα shell.

For transmission electron microscopy (TEM) images of fluorapatite particles and hydrogels with FAp particles, samples were immobilized on Pioloform-coated 100-mesh copper grids (Plano GmbH, Wetzlar, Germany) and stained with Uranyl acetate. JEM-2100 TEM (JEOL, Akishima, Japan) was operated at 80 kV, and images were taken using a 4000 × 4000 pixels charge-coupled device camera (UltraScan 4000, Gatan, Pleasanton, CA, USA) and Gatan Digital Micrograph software (version 1.83.842). Particle size was determined from 10 individual particles using ImageJ software (NHI, Bethesda, MD, USA).

Light microscopy images of cells and hydrogels were recorded in a wet state using a Leica DM IL LED microscope (Leica, Wetzlar, Germany) and processed using LAS 4.8 software (Leica, Germany).

Confocal laser scanning microscopy was carried out in wet state using a Leica CLSM TCS SP8 (Leica, Germany) and LAS software; images were processed using ImageJ (NHI, Bethesda, MD, USA).

### 2.8. Rheology

Rheological data were recorded as triplicate using a Discovery Hybrid Rheometer 3 (TA Instruments, New Castle, DE, USA) with a plate-plate geometry (diameter 25 mm) and a sample volume of 500 µL (n = 3), and a gap size of 500 µm at 25 °C. A wet sponge adapter around the geometry served to prevent the premature drying of the hydrogels. Frequency sweep experiments were recorded at angular frequencies between 0.1–100 rad/s and 100–0.1 rad/s for recovery at 50% strain. Amplitude sweeps were recorded at 31.4 rad/s and a strain of 0.1–1000%. These settings were used in previous studies to examine eADF4(C16)-hydrogel characteristics.

### 2.9. Dynamic Light Scattering

Dynamic light scattering (DLS) was measured using a Litesizer^TM^ 500 (Anton Paar, Graz, Austria). Diluted particle samples were recorded in 10 mM Tris buffer, pH 7.5, in omega cuvettes at 25 °C. Mean values of zeta potentials were automatically calculated from internal spectra using the Kalliope software and the Smoluchowski model [44].

### 2.10. Cell Culture

Cytotoxicity was analyzed according to DIN EN ISO 10993-5 using BALB/3T3 mouse fibroblasts. BALB/3T3 mouse fibroblasts (ACC210, ATCC, Manassas, VA, USA) were subsequently cultured in Dulbecco’s Modified Eagle Media (Biochrom, Germany) with supplemented 10% *v*/*v* fetal calve serum (Bio&Sell, Feucht, Germany), 1% *v*/*v* GlutaMax (Invitrogen, Waltham, MA, USA) and 0.1% *v*/*v* gentamycin sulfate (Sigma-Aldrich, St. Louis, MO, USA) at 37 °C, 5% CO_2_ and 95% relative humidity in a cell culture incubator (HERAcell 150i, Thermo Fisher Scientific, Waltham, MA, USA). Sub-confluent cultures were seeded at 25,000 cells/cm^2^ on treated tissue culture plates (TCP, Nunclon, Thermo Fisher Scientific, Waltham, MA, USA) 24 h prior to the test (passage 9). Particles, high-density polyethylene, and organotin-stabilized polyurethane were UV-treated for 30 min prior to use (n = 3). For extraction tests, 10 mg of fluorapatite or eADF4(κ16)-coated fluorapatite particles were incubated in 1 mL cell culture media for 24 h at 37 °C. For direct contact tests, between 10 and 80% of the test area was covered with the particles. To quantify cell viability, 10% *v*/*v* CellTiter-Blue reagent (Promega, Madison, WI, USA) was incubated on washed cells for 2.5 h. Then, 100 µL of the sample was analysed concerning resazurin to resofurin transformation at 590 nm using a plate reader (Mithras LB 940, Berthold Technologies, Bad Wildbad, Germany). Significance was calculated using ANOVA statistic in Origin (Northampton, MA, USA) in a Tukey test with *p* < 0.05. Images were collected directly after the test.

For gradient bioprinting of cells, 10^6^ cells/mL were encapsulated (at passage 10) with 15% cell culture media in 3% *w*/*v* spider silk pre-gel solutions complemented with 1% *w*/*v* eADF4(κ16)-coated FAp particles and gelled at 37 °C. Cell viability was confirmed using trypan blue (Sigma-Aldrich, St. Louis, MO, USA) in a Cell Titer Blue Assay and an automated cell counter (TC20, Bio-Rad, Hercules, CA, USA). Cells were live-dead stained with calcein acetoxymethylester (abbreviation: calcein AM) and ethidium homodimer I (Invitrogen, Thermo Fisher Scientific, Waltham, MA, USA) for 45 min before imaging.

## 3. Results and Discussion

Bioprinting depends on parameters such as extrusion velocity, printing speed, strand thickness by nozzle diameter, but also rheological properties of the used materials. There are different types of printing, and here the focus was on pneumatic extrusion printing of hydrogels. A new set-up was established to process two separate materials out of one printer cartridge. The aim was to generate scaffolds with gradient properties without strand breakup.

### 3.1. Flow Simulation of Gradient Material Printing

When two materials are present in one printer cartridge, a back mixing effect can be seen both in flow simulations and in printing experiments. This effect was related to the U-shaped profile of a laminar material flow, creating a velocity gradient. With the exemplary extrusion velocity of already 5 mm/s, a backmixing of two materials could be shown in a cartridge model, creating a gradient profile in the extruded strand (Figure 1b).

The concentration curve for both fluids at the cartridge outlet is shown in Figure 1a (additional Appendix A). When running the simulation with no slip conditions assumed at the cartridge walls, a core-shell effect of both materials was found (Figure 1c) in addition to mixing, as reported earlier [45]. Furthermore, the residual material A at the cartridge walls was extruded after the piston met the piston crown.

### 3.2. Experimental Confirmation of Gradient Formation after Bioprinting

A commercial 3cc printer cartridge providing a conical piston, and a 20 G conical needle was used in a regenHU 3D Discovery printer with a printing speed of 20 mm/s and 0.1 bar. Two different materials, 0.5 mL each, were filled into one cartridge. First, colored and uncolored face creams were tested for visualization (Appendix A). Then, hydrogels made of 3% *w*/*v* eADF4(C16) [31] were prepared as material block A. As block B, a fluorescent 3% *w*/*v* eADF4(C16) hydrogel labeled with 10% *w*/*w* of FITC-eADF4 (C16) was used. Only a slight decrease in viscosity and stiffness was observed in rheological measurements in the case of the eADF4(C16)/FITC-eADF4(C16) blend hydrogel (Appendix A), which was in accordance with previous observations [31]. A visual colorization from material B at one side of the printed construct to a transparent hydrogel of material A was obtained after printing (Figure 2a).

Fluorescent imaging of the construct showed a gradual increase in fluorescence signal in accordance with the increase in FITC-eADF4(C16) content (Figure 2b). Small amounts of the hydrogel were retracted from the construct at distinct locations for quantified fluorescence analysis. From unlabeled to fluorescence-labeled hydrogels, an increase in fluorescence intensity was measured using three replicate scaffolds (Figure 2c). This profile perfectly fitted the simulated material profile and confirmed the accordance of the simulation with the experimental set-up. However, after reaching maximum fluorescence, a plateau phase and finally an intensity decrease were recorded. This indicates the occurrence of an inner filter effect, typically observed at high concentrations of fluorophores [46].

### 3.3. Spider Silk Fluorapatite Composite Hydrogels to Produce Bioactive Gradients

The tendon-to-bone interface shows a gradient in mechanical, compositional, and structural cues, and, therefore, the applicability of the presented system was tested in biofabricating such an interface. First, fluorapatite particles were gradually integrated to mimic apatite mineralization towards the bone side.

Fluorapatite (FAp) particles were synthesized using an ultrasonication approach from dry components as previously reported by Willigeroth [43], yielding rod-shaped FAp particles with 92 ± 27 nm in length and 21 ± 6 nm width as visualized using SEM and TEM (Appendix A). The chemical integrity of the particles was identified in comparison to the quantitative occurrence of the elements Ca, F, O, and P in FAp stoichiometry using SEM-EDX for element analysis (Appendix A). The variations between calculated and experimentally determined ratios for Ca:F and Ca:P was 2%, and for the Ca:O ratio, 21%. This results in a higher O-content than expected, which could be explained by atmospheric or intracrystalline water. Furthermore, ATR-FTIR spectra of FAp particles were compared to that of commercially available hydroxyapatite particles (Appendix A). Both materials showed similar band assignments as previously reported for both apatite species [47].

The implementation of fluorapatite particles into spider silk solutions before gelation enabled homogenous incorporation into the generated hydrogels as detected using TEM (Figure 3a), showing intertwined FAp particles with eADF4(C16) fibrils. Concerning particle content, an increase from 1% to 3% *w*/*v* yielded a higher initial storage modulus of the hydrogels (Figure 3b). At higher strains, severe differences were visible, as one material started to flow whilst the other was still static.

For simultaneous printing of blank and particle-filled hydrogels from one cartridge, it was crucial that both materials had flow points in the same order of magnitude. Therefore, a material blend of 3% *w*/*v* eADF4(C16) and 1% *w*/*v* FAp or κFAp particles was used further on.

One important property of eADF4(C16) hydrogels is their shear thinning behavior [31,33,34,37]. In the presence of FAp particles, the hydrogels showed shear-thinning behavior known for non-Newtonian fluids at increasing shear rates and recovery at decreasing shear rates, which is important to gain solid structures after strand deposition upon printing (Figure 3c). Similar behavior was found for recombinant silk particle-enforced [35] and silica particle-enforced [36] hydrogels.

Gradient printing of recombinant spider silk hydrogels in the presence of FAp particles (material A) and in their absence (material B) was realized (Figure 4a) with the identical printing parameters as used for single-material hydrogels. There was a slight dewetting effect on the substrate, visible between positions II and IV (Figure 4a), without influencing the general outcome. Light microscopy images at distinct positions revealed a decreasing particle density in the hydrogel (Figure 4b, I–VI). Small particle aggregates in the lower micrometer range were visible, especially in regions with higher particle concentrations due to particle aggregation during extrusion through the nozzle.

### 3.4. Biofabrication of Particle and Cell-Loaded Recombinant Spider Silk Hydrogels

As FAp particles showed a negative zeta potential (−22.5 ± 0.9 mV), the interaction of the particles with negatively charged eADF4(C16) in solution could be increased upon coating FAp particles with the positively charged spider silk variant eADF4(κ16), referred to as κFAp (zeta potential + 16.5 ± 0.4 mV). The rheological behavior of the blended hydrogels was not significantly influenced (Figure 3b,c).

To use composite hydrogels for biofabrication, first, the incorporated particles were individually tested regarding cell toxicity according to DIN EN ISO 10993-5. An extract test and a direct contact test were carried out. A cytotoxic effect is considered when cell viability is reduced by 30%, referred to as high-density polyethylene used as a positive control. As a negative control, organotin-stabilized polyurethane was used. Cell viability decreased in the presence of fluorapatite particles in the direct contact test to 15.7 ± 8.1% and to 34.8 ± 2.4% in the extract test. Particles adhered to the cells and could hardly be washed off (Appendix A).

Materials made of recombinant spider silk proteins have already been confirmed earlier as biocompatible materials, showing no immune reaction or cytotoxicity [48,49]. eADF4(κ16)-coating of fluorapatite particles enhanced the cell viability in both the extract and contact test significantly (Appendix A). Higher cell viability of 65.9 ± 4.4% could be detected in the direct contact test and 57.8 ± 10.6% in the extract test. Besides these enhancing effects on cell viability, a silk-coating could not shift cytotoxicity to less than 30%, referred to as the positive control. However, cell toxic effects of apatite particles on cells had already been reported in the literature, depending on particle shape as well as cell line [50].

Additionally, for biofabrication, as referring to the DIN test, BALB/3T3 mouse fibroblasts (10^6^ cells/mL) were encapsulated in 3% *w*/*v* eADF4(C16) hydrogels together with 1% *w*/*v* κFAp particles. The AB block system comprised 0.5 mL each of 3% *w*/*v* eADF4(C16) + 1% *w*/*v* κFAp + fibroblasts (material A) and 3% *w*/*v* eADF4(C16) hydrogel (material B). Bioprinting was carried out using a 16 G conical needle. Hydrogels with cells [37] and particles were slightly stiffer than the non-cell-loaded ones. The obtained scaffolds were stained using calcein AM/ethidium homodimer I for live-dead evaluation of the cells. Spider silk hydrogels showed slight red autofluorescence, as seen previously [33]. The confocal laser scanning microscopy images at distinct positions on the scaffold confirmed successful gradient bioprinting of BALB/3T3 fibroblasts (Figure 5).

The simultaneous processing of apatite particles with a cell-friendly protein coating along with BALB/3T3 cells showed that this printing set-up offers not only the possibility to generate gradients from one printer cartridge but also to incorporate multiple gradient features into one scaffold (here, particles and cells), relevant for, e.g., biofabrication of tissues at the tendon-bone interface [51]. Various additional features and fillers could be added in future studies (Figure 6).

## 4. Conclusions

Computational fluid dynamics combined with bioprinting enabled to achieve an in situ generated gradient construct from one printer cartridge. The combined numerical and experimental approach was used to generate gradient constructs from two different materials. The focus was set on the evaluation of the bioprinting process for gradient generation. First, computational fluid dynamics allowed the prediction of the gradient using a transient laminar flow simulation for an AB material block system. Typical shear-thinning material behavior using Ostwald de Waele’s power law was taken into account. This new AB block set-up could then be applied for various material combinations and complexity levels, and mineralization gradients or even mineralization gradients together with cell gradients could be produced. The combination of inorganic fillers and cells simultaneously in a material gradient can yield similar conditions as found at the enthesis. Upon material optimization according to the mechanical properties of the natural tissue, future applications in biofabricating tissues for the tendon-to-bone interface could be realized utilizing this novel printing method.

## Figures and Tables

**Figure 1 biomolecules-12-01413-f001:**
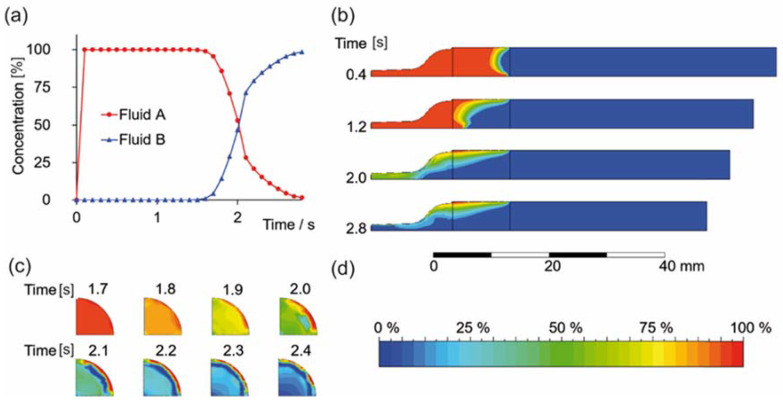
Flow simulation results. (**a**) Concentration of fluid A and B of an AB block system in a printing cartridge at the outlet during the extrusion process. (**b**) The concentration profile within the cartridge during extrusion shows a back mixing effect in the cartridge clip plane. (**c**) Occurring core-shell effect before the final material extrusion of fluid A at the outlet (cross section). (**d**) Color scale for (**b**) and (**c**), depicting fluid A in red and fluid B in blue (blue means 0% fluid A).

**Figure 2 biomolecules-12-01413-f002:**
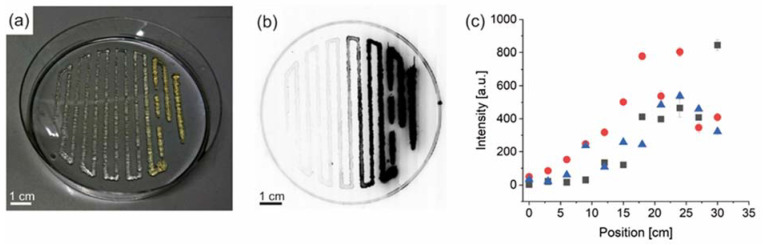
Three-dimensional printed gradient of recombinant spider silk hydrogels (3% *w*/*v*) with one spider silk component fluorescently labeled. (**a**) Photograph of gradient hydrogel constructs from non-labeled (left) to fluorescently labeled spider silk proteins (yellow, right). (**b**) Gradient fluorescence signal visualization of the same sample as in a. (**c**) Quantification of the fluorescence signal at distinct locations along the printed strand from non-labeled (left) to labeled (right) spider silk proteins; red, blue, and black symbols represent measurements for three identical scaffolds as statistic repeats.

**Figure 3 biomolecules-12-01413-f003:**
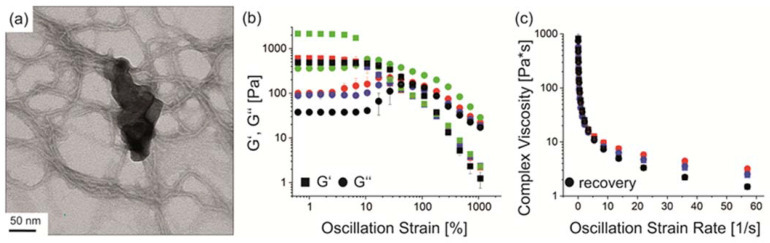
Fluorapatite particle (FAp) with eADF4(C16) hydrogel composites. (**a**) TEM image of eADF4(C16) fibrils and intertwined FAp particles. (**b**) Mean rheological amplitude sweep measurements of 3% *w*/*v* eADF4(C16) (black), 3% *w*/*v* eADF4(C16) + 1% *w*/*v* FAp (red), 3% *w*/*v* eADF4(C16) + 1% *w*/*v* κFAp (blue) and 3% *w*/*v* eADF4(C16) + 3% *w*/*v* FAp (green) composite hydrogels. (**c**) Mean rheological frequency sweep measurements of 3% *w*/*v* eADF4(C16) (black), 3% *w*/*v* eADF4(C16) + 1% *w*/*v* FAp (red) and 3% *w*/*v* eADF4(C16) + 1% *w*/*v* κFAp (blue) composite hydrogels. κFAp refers to FAp particles coated with eADF4(κ16).

**Figure 4 biomolecules-12-01413-f004:**
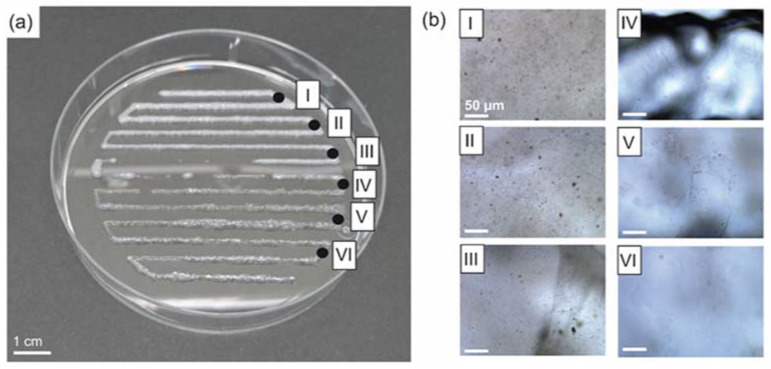
Three-dimensional printed gradient of recombinant spider silk hydrogels (3% *w*/*v*) with and without fluorapatite particles. (FAp) (**a**) Gradient from 3% *w*/*v* eADF4(C16) + 1% *w*/*v* κFAp (white) to 3% *w*/*v* eADF4(C16) (transparent). Numbers I-VI indicate the positions for enlarged images in (**b**). (**b**) Light microscopy images at distinct locations of the construct showing decreasing particle concentration (highest at position I and lowest at position VI), scale bar 50 µm. κFAp refers to FAp particles coated with eADF4(κ16).

**Figure 5 biomolecules-12-01413-f005:**
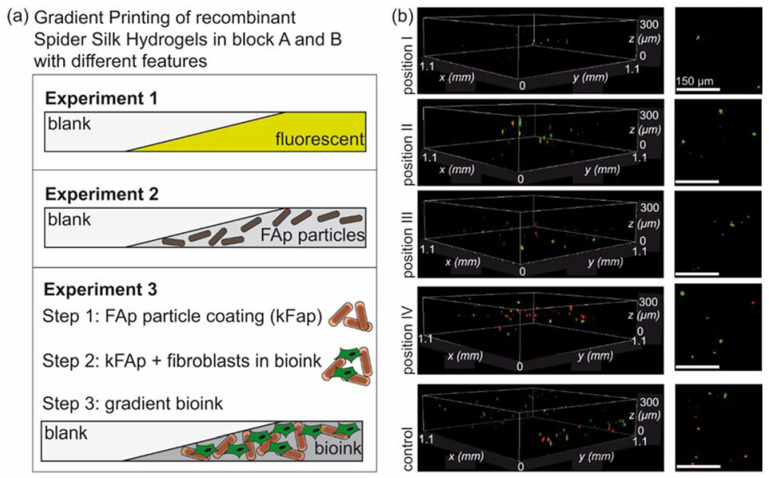
(**a**) Scheme of different experimental set-ups for gradient printing from one printer cartridge with different features. (**b**) Confocal laser scanning microscopy images of fluorescently labeled BALB/3T3 fibroblasts using ethidium homodimer I (red: dead cells)/calcein AM (green: living cells) staining. Cells and 1% *w*/*v* κFAp particles were loaded into a hydrogel and gradient-printed towards an unloaded hydrogel (both hydrogels 3% *w*/*v* eADF4(C16)); position I with lowest and position III with highest concentration of cells/particles. Static cell and particle-loaded hydrogels served as control. Left: side view, right: top view. Scale bars right row 150 µm. κFAp refers to FAp particles coated with eADF4(κ16).

**Figure 6 biomolecules-12-01413-f006:**
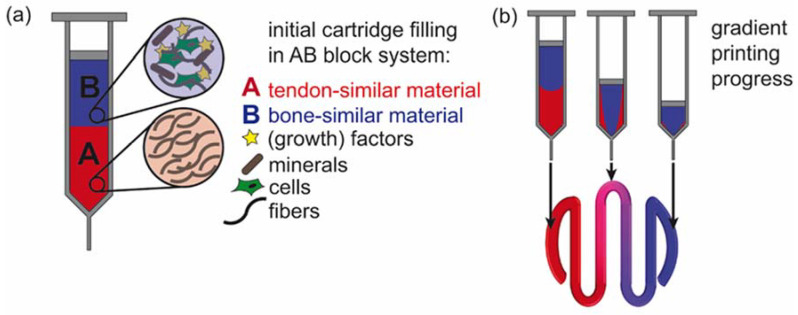
Concept of gradient printing from one printer cartridge of materials relevant for biofabrication of tissues at the tendon/bone interface. (**a**) AB block filling of a tendon-similar material (A, red) and a bone-similar material (B, blue) with tissue-specific features. (**b**) Material extrusion during printing and gradient scaffold generation.

## Data Availability

The data that support the findings of this study are available from the corresponding author upon reasonable request.

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
