# Peer review of "Flow Simulation and Gradient Printing of Fluorapatite- and Cell-Loaded Recombinant Spider Silk Hydrogels"

_biomolecules, 2022, doi:10.3390/biom12101413_

Round 1

Reviewer 1 Report

This work reports a printing method to generate gradient structures from one printer cartridge. Both the mineralization gradients and the cell gradients were achieved together. The combination of computational and experimental approaches illustrates the systematical exploration. The results and figures are well organized and presented clearly. However, there are a few issues should be addressed.

1.      The authors target to the application of tendon-to-bone-interface with a gradient of mechanical properties. However, the mechanical properties of printed hydrogels were not directly measured. The stress-strain curves or compression strength curves should be included to demonstrate the successful control of properties.

2.      From the photos, it seems that the uniformity of line width and continuity of the printed lines were not well controlled. Will the uniformity and the continuity influence the performances in potential applications?

3.      Can the printed structure maintain the gradients for long time? Please provide the stability results for both the mineralization gradients and the cell gradients

4.      How are the cell viability and proliferation activity in the printed structure?

Author Response

We thank the reviewer for the positive evaluation.

Reviewer 2 Report

I have reviewed the manuscript entitled"Flow Simulation and Gradient Printing of Fluorapatite and Cell loaded Recombinant Spider Silk Hydrogels".In this manuscript, combined numerical and experimental methods were employed to fabricate a gradient construct using two different materials. I found this manuscript very interesting and well-written. The manuscript has provided concrete evidence for all claims and I think this manuscript can be considered for publication in the present form. 

Author Response

This work reports a printing method to generate gradient structures from one printer cartridge. Both the mineralization gradients and the cell gradients were achieved together. The combination of computational and experimental approaches illustrates the systematical exploration. The results and figures are well organized and presented clearly. However, there are a few issues should be addressed.

  1. The authors target to the application of tendon-to-bone-interface with a gradient of mechanical properties. However, the mechanical properties of printed hydrogels were not directly measured. The stress-strain curves or compression strength curves should be included to demonstrate the successful control of properties.

In the experimental frame of the study, mechanical properties of particle loaded hydrogels were determined in rheological measurements as depicted in Figure 3. The aim was to confirm shear-thinning behaviour of the hydrogels as a prerequisite for printing experiments. For future applications with tailored mechanical properties as for the tendon-to-bone-interface, far higher particle and cell loading densities would be necessary. In a first place, the focus was on the gradient generation as a novel process as a first mandatory step towards the generation of a suitable replacement material. To come close to mechanical properties of a natural tissue such as the enthesis, the bioink will have to be improved. According explanations were added to the manuscript to clarify this issue.

  1. From the photos, it seems that the uniformity of line width and continuity of the printed lines were not well controlled. Will the uniformity and the continuity influence the performances in potential applications?

The photos referred to in Figure 2 and 4 show large Petri dishes with very thick lines. The high line width was chosen to make the gradient macroscopically visible (see also scale bar in Figure 2a+4a). More sophisticated printed structures in 3D could be achieved with unloaded hydrogels from eADF4(C16) spider silk protein. Previous studies cited in the manuscript showed the 3D bioprinting of unloaded hydrogels (DeSimone, E.; Schacht, K.; Pellert, A.; Scheibel, T. Recombinant spider silk-based bioinks. Biofabrication 2017, 9 (4), 044104.) and high precision processing of heart-valve-like structures (Lechner, A., Trossmann, V.T. and Scheibel, T. Impact of Cell Loading of Recombinant Spider Silk Based Bioinks on Gelation and Printability. Macromol. Biosci., 2022, 22: 2100390.), particle-loaded hydrogels (Kumari, S.; Bargel, H.; Scheibel, T. Recombinant Spider Silk–Silica Hybrid Scaffolds with Drug-Releasing Properties for Tissue Engineering Applications. Macromol. Rapid Commun. 2020, 41 (1), 1900426.) and with variable material composition (Neubauer, V. J.; Trossmann, V. T.; Jacobi, S.; Döbl, A.; Scheibel, T. Recombinant Spider Silk Gels Derived from Aqueous–Organic Solvents as Depots for Drugs. Angew. Chem.-Int. Edit. 2021, 60 (21), 11847-11851.). The material has been confirmed to allow low line width and 3D geometries, which could potentially be chosen for each single potential application accordingly. See also the exclusive figure for the response letter below with unpublished photos showing a 3D spider silk construct from a eADF4(C16) blend hydrogel printed with a 22G needle (inner diameter 0.41 mm).

  1. Can the printed structure maintain the gradients for long time? Please provide the stability results for both the mineralization gradients and the cell gradients.

Degradation studies have already been conducted in previous studies, showing a long-term stability of eADF4(C16) printed constructs with incorporated silica particles and BALB3T3 fibroblasts over minimum two weeks. See: Kumari, S.; Bargel, H.; Scheibel, T., Recombinant Spider Silk–Silica Hybrid Scaffolds with Drug-Releasing Properties for Tissue Engineering Applications. Macromol. Rapid Commun. 2020, 41 (1), 1900426.

  1. How are the cell viability and proliferation activity in the printed structure?

Viability studies were previously conducted in printed eADF4(C16) hydrogels directly after printing and after 7 and 14 days of subsequent cultivation as shown in Lechner, A., Trossmann, V.T. and Scheibel, T. Impact of Cell Loading of Recombinant Spider Silk Based Bioinks on Gelation and Printability. Macromol. Biosci., 2022, 22: 2100390.). See Figure 5 from this publication:

Reviewer 3 Report

The paper focuses on a novel method of bioprinting to generate gradient biomaterial constructs. Numerical simulations were provided along with a simple setup of gradient bioprinting. Understanding the transition between different tissues with different mechanical properties is important in the field. Therefore, the study should be considered of value. I am providing some edits and suggestions below which will improve the quality of the paper.

1- The explanation of production steps starts with “bioprinting”, then goes on with “hydrogel preparation and particle synthesis”. However, it would be more appropriate to explain firstly the hydrogel and particle synthesis and then the bioprinting process.

2- Lines 60-66: Fluorapatite properties are given and stated that FAp was successfully used for osteoporosis treatment and bone tissue engineering. So, for cell studies of gradient scaffold, osteoblast cells could be used instead of fibroblast cells to improve the regeneration. What are the superior properties of fibroblast cells instead of osteoblast cells to use with FAp?

3- Discuss why fluorapatite was preferred over fluoride and explain the relationship between fluoride and fluorapatite.

4- Lines 69-71: Insufficient information is given about computational fluid dynamics. This technique should be given in more detail.

5- Line 75: Fix the typo: "gelatine"

6- Does the scaffold occur from spider silk hydrogel or spider silk protein hydrogel? Also, what is the beneficial effect of using spider silk? Why was this hydrogel preferred?

7- This expression “eADF4(κ16)” is given with different “k” in lines 197, 207 and 332. It should be rewritten in correct form.

8- Line 92: Why the liquids' length ratio was chosen as 1:2. What is the benefit in using different length ratio?

9- Lines 140-141: Full names of the chemicals “CaCl2, Na2HPO4 and NaF” were not given.

10- Line 144: Replace “over night” with “overnight”.

11- Line 208: Trypan blue was used in the cell culture part, but it has not been shown to be used with any data. In figure S5, it is stated that titer blue assay is used. Please recheck.

12- Below the caption “2.10. Cell culture”, steps of the cell viability, live/dead, cytotoxicity assays steps should be given in detail. There is only general information about these assays in the article.

13- Explain what are the extract test and direct contact test?

14- Lines 259-261: It is stated that there are 3 individual scaffolds. What are these scaffolds, were all of them printed with the same hydrogel?

15- In the Figure 3 caption, groups of hydrogels were mentioned at b) as 3% w/v eADF4(C16) (black), 3 % w/v eADF4(C16) +1% w/v FAp (red), 3% w/v eADF4(C16)+1% w/v 280 kFAp (blue) and 3% w/v eADF4(C16)+3% w/v FAp (green) composite hydrogels. One group contains kFAp and others contain FAp, however the difference between kFAp and FAp were not given. What is the difference of kFAp and purpose of using kFAp?

16- Figure 3: In c), groups of hydrogel mentioned as 3% w/v eADF4(C16) (black), 3% w/v eADF4(C16)+1 %  w/v FAp (red) and 3% w/v eADF4(C16)+1 % w/v kFAp (blue). Why there is no data about 3% w/v eADF4(C16)+3% w/v FAp (green) composite?

17- Lines 326, 327: It is stated that polyethylene was used as positive control group and polyurethane as negative control group. What is the purpose of choosing these polymers as positive and negative control groups?

18- Line 327: Fix the typo, “viablity”

19- Lines 344-346: It is stated that for biofabrication 3 % w/v eADF4(C16)+1% w/v kFAp+fibroblasts blend was used. However, in lines 291-292, it is stated that 3 % w/v eADF4(C16) and 1 % w/v FAp blend was used due to the same flow point of blank and particle filled hydrogel. So, why kFAp is used instead of FAp for bioprinting?

20- Lines 350-352: The explanation of Figure 5 is not enough. It should be explained in detail. In addition, in the figure 5 caption, there is no “(a)” and “(b)”.

21- Cell viability test; how long after the start of the experiments, images were collected?

22- Figure S4, subfigure (d): typo in y-axis

23- Conclusion should be expanded.

Author Response

The paper focuses on a novel method of bioprinting to generate gradient biomaterial constructs. Numerical simulations were provided along with a simple setup of gradient bioprinting. Understanding the transition between different tissues with different mechanical properties is important in the field. Therefore, the study should be considered of value. I am providing some edits and suggestions below which will improve the quality of the paper.

  • The explanation of production steps starts with “bioprinting”, then goes on with “hydrogel preparation and particle synthesis”. However, it would be more appropriate to explain firstly the hydrogel and particle synthesis and then the bioprinting process.

The presentation of the experimental results was chosen in this order as the focus of the studies was set on the bioprinting process, not the hydrogel preparation or particle synthesis. In the very first experiment, coloured and uncoloured face cream were printed for visualization (Figure SI 2) (see line 251), making it necessary to explain the novel printing process first.

  • Lines 60-66: Fluorapatite properties are given and stated that FAp was successfully used for osteoporosis treatment and bone tissue engineering. So, for cell studies of gradient scaffold, osteoblast cells could be used instead of fibroblast cells to improve the regeneration. What are the superior properties of fibroblast cells instead of osteoblast cells to use with FAp?

BALB/3T3 fibroblasts were chosen first to evaluate cytotoxicity as required in the DIN EN ISO 10993-5 test for FAp particle evaluation and second to demonstrate cell viability during and after the printing process, again with the same cell line. The choice of this cell line is now clarified in line 362.

  • Discuss why fluorapatite was preferred over fluoride and explain the relationship between fluoride and fluorapatite.

Fluorapatite (Ca5(PO4)3F) is chemically related to hydroxyapatite (Ca5(PO4)3OH), which can be found in the natural bone and can be used as particles for tissue engineering. Fluoride is F-, the salt of hydrofluoric acid, and is not appropriate for tissue engineering in this from. However, fluoride is an anionic component of fluorapatite.

  • Lines 69-71: Insufficient information is given about computational fluid dynamics. This technique should be given in more detail.

We thank the reviewer for this hint and adopted the manuscript accordingly (see line 73 ff): “Computational fluid dynamics allowed the prediction of gradient formation using a transient laminar flow simulation in a commercial printer cartridge filled with an AB material block system taking into account the typical shear-thinning material behaviour using Ostwald de Waele's power law.”

  • Line 75: Fix the typo: "gelatine"

Corrected accordingly

  • Does the scaffold occur from spider silk hydrogel or spider silk protein hydrogel? Also, what is the beneficial effect of using spider silk? Why was this hydrogel preferred?

Spider silk hydrogel is used as synonym for spider silk protein hydrogel. Spider silk protein hydrogels such as from engineered spider silk proteins can be used for biomedical applications and such materials show outstanding properties with regard to cell-surface adherence, mechanical and textural properties, no toxicity, low / no immunogenicity and biodegradability of scaffolds compared to other materials (see: Leal-Egana, A.; Scheibel, T., Silk-based materials for biomedical applications. Biotechnol. Appl. Biochem. 2010, 55, 155-167.)

  • This expression “eADF4(κ16)” is given with different “k” in lines 197, 207 and 332. It should be rewritten in correct form.

Corrected accordingly

  • Line 92: Why the liquids' length ratio was chosen as 1:2. What is the benefit in using different length ratio?

We thank the reviewer for this hint and adopted the manuscript accordingly (see line 97 ff): “The initial ratio of the lengths of blocks A and B was 1:2 to ensure a flow of almost only material B at the end of the gradual mixing process. Furthermore, this ensured that the mixing area was located far away from the piston, so that the piston shape did not influence the mixing.”

  • Lines 140-141: Full names of the chemicals “CaCl2, Na2HPO4 and NaF” were not given.

Corrected accordingly

  • Line 144: Replace “over night” with “overnight”.

Corrected accordingly

  • Line 208: Trypan blue was used in the cell culture part, but it has not been shown to be used with any data. In figure S5, it is stated that titer blue assay is used. Please recheck.

Trypan blue was used for the Cell Titer Blue Assay. According quantitative data are shown in Figure S5 a.

  • Below the caption “2.10. Cell culture”, steps of the cell viability, live/dead, cytotoxicity assays steps should be given in detail. There is only general information about these assays in the article.

All used chemicals and procedures are listed in 2.10. Detailed information is also given for the DIN EN ISO 10993-5 test, at line 200 ff.

  • Explain what are the extract test and direct contact test?

Both tests are part of the DIN EN ISO 10993-5 procedure and explained at line 200 ff.

  • Lines 259-261: It is stated that there are 3 individual scaffolds. What are these scaffolds, were all of them printed with the same hydrogel?

Yes, the three scaffolds were replicates from the same material and batch.

  • In the Figure 3 caption, groups of hydrogels were mentioned at b) as 3% w/v eADF4(C16) (black), 3 % w/v eADF4(C16) +1% w/v FAp (red), 3% w/v eADF4(C16)+1% w/v 280 kFAp (blue) and 3% w/v eADF4(C16)+3% w/v FAp (green) composite hydrogels. One group contains kFAp and others contain FAp, however the difference between kFAp and FAp were not given. What is the difference of kFAp and purpose of using kFAp?

FAp particles were treated in aqueous solutions of eADF4(k16) protein to generate a protein coating. For the process, see lines 155. The coating is now mentioned in this caption to clarify, also in captions to Figure 4, 5, S5 and the materials&methods part.

  • Figure 3: In c), groups of hydrogel mentioned as 3% w/v eADF4(C16) (black), 3% w/v eADF4(C16)+1 % w/v FAp (red) and 3% w/v eADF4(C16)+1 % w/v kFAp (blue). Why there is no data about 3% w/v eADF4(C16)+3% w/v FAp (green) composite?

Please see line 304: “At higher strains, severe differences were visible, as one material started to flow whilst the other was still static. For simultaneous printing of blank and particle-filled hydrogels from one cartridge it was crucial that both materials had flow points in the same order of magnitude. Therefore, a material blend of 3 % w/v eADF4(C16) and 1 % w/v FAp particles was used further on. 3% w/v eADF4(C16)+3% w/v FAp (green) was therefore not tested for shear-thinning behaviour.”

  • Lines 326, 327: It is stated that polyethylene was used as positive control group and polyurethane as negative control group. What is the purpose of choosing these polymers as positive and negative control groups?

Both materials are defined as controls by the applied DIN EN ISO 10993-5 procedure.

  • Line 327: Fix the typo, “viablity”

Corrected accordingly

  • Lines 344-346: It is stated that for biofabrication 3 % w/v eADF4(C16)+1% w/v kFAp+fibroblasts blend was used. However, in lines 291-292, it is stated that 3 % w/v eADF4(C16) and 1 % w/v FAp blend was used due to the same flow point of blank and particle filled hydrogel. So, why kFAp is used instead of FAp for bioprinting?

Blends with 1 % FAp or kFAp were used, since the protein coating was advantageous in contrast to non-coated particles. To clarify, the sentence in line 308 was corrected.

  • Lines 350-352: The explanation of Figure 5 is not enough. It should be explained in detail. In addition, in the figure 5 caption, there is no “(a)” and “(b)”.

“(a)” and “(b)” were added. The explanation for section a) was added: “. a) Scheme of different experimental setups for gradient printing from one printer cartridge with different features.”

  • Cell viability test; how long after the start of the experiments, images were collected?

See line 211: “To quantify cell viability, 10 % v/v CellTiter-Blue reagent (Promega, Germany) were incubated on washed cells for 2.5 h” and line 216: “Images were collected directly after the test.”

  • Figure S4, subfigure (d): typo in y-axis

Corrected accordingly

23 Conclusion should be expanded.

The conclusion was expanded from line 384: “The combined numerical and experimental approach was used to generate gradient constructs from two different materials. The focus was set on the evaluation of the bioprinting process for gradient generation. First, computational fluid dynamics allowed the prediction of the gradient using a transient laminar flow simulation for a AB material block system. Typical shear-thinning material behaviour using Ostwald de Waele's power law was taken into account. This new AB block set-up could then be applied for various material combinations and complexity levels, and mineralization gradients or even mineralization gradients together with cell gradients could be produced. The combination of inorganic fillers and cells simultaneously in a material gradient can yield similar conditions as found at the enthesis. Upon material optimization according to the mechanical properties of the natural tissue, future applications in biofabricating tissues for the tendon-to-bone interface could be realized utilizing this novel printing method.”